# Dietary Oat Bran Reduces Systemic Inflammation in Mice Subjected to Pelvic Irradiation

**DOI:** 10.3390/nu12082172

**Published:** 2020-07-22

**Authors:** Piyush Patel, Dilip Kumar Malipatlolla, Sravani Devarakonda, Cecilia Bull, Ana Rascón, Margareta Nyman, Andrea Stringer, Valentina Tremaroli, Gunnar Steineck, Fei Sjöberg

**Affiliations:** 1Division of Clinical Cancer Epidemiology, Department of Oncology, Institute of Clinical Sciences, The Sahlgrenska Academy, University of Gothenburg, 41345 Gothenburg, Sweden; dilip.kumar.malipatlolla@gu.se (D.K.M.); sravani.devarakonda@gu.se (S.D.); cecilia.bull@gu.se (C.B.); gunnar.steineck@oncology.gu.se (G.S.); fei.sjoberg@microbio.gu.se (F.S.); 2Department of Infectious Diseases, Institute of Biomedicine, the Sahlgrenska Academy, University of Gothenburg, 41346 Gothenburg, Sweden; 3Department of Food Technology, Engineering and Nutrition, Lund University, 22100 Lund, Sweden; Ana.Rascon@aventureab.com (A.R.); margareta.nyman@food.lth.se (M.N.); 4School of Pharmacy and Medical Sciences, University of South Australia, Adelaide SA 5001, Australia; andrea.stringer@unisa.edu.au; 5The Wallenberg Laboratory, Department of Molecular and Clinical Medicine, the Sahlgrenska Academy, University of Gothenburg, 413 45 Gothenburg, Sweden; Valentina.Tremaroli@wlab.gu.se

**Keywords:** pelvic radiotherapy, radiation-induced inflammation, serum cytokines, dietary fiber, oat bran

## Abstract

Patients undergoing radiotherapy to treat pelvic-organ cancer are commonly advised to follow a restricted fiber diet. However, reducing dietary fiber may promote gastrointestinal inflammation, eventually leading to deteriorated intestinal health. The goal of this study was to evaluate the influence of dietary fiber on radiation-induced inflammation. C57BL/6J male mice were fed a High-oat bran diet (15% fiber) or a No-fiber diet (0% fiber) and were either irradiated (32 Gy delivered in four fractions) to the colorectal region or only sedated (controls). The dietary intervention started at 2 weeks before irradiation and lasted for 1, 6, and 18 weeks after irradiation, at which time points mice were sacrificed and their serum samples were assayed for 23 cytokines and chemokines. Our analyses show that irradiation increased the serum cytokine levels at all the time points analyzed. The No-fiber irradiated mice had significantly higher levels of pro-inflammatory cytokines than the High-oat irradiated mice at all time points. The results indicate that a fiber-rich oat bran diet reduces the intensity of radiation-induced inflammation, both at an early and late stage. Based on the results, it seems that the advice to follow a low-fiber diet during radiotherapy may increase the risk of decreased intestinal health in cancer survivors.

## 1. Introduction

Radiotherapy plays an important role in cancer treatments with curative or palliative intent. In many countries, more than half of cancer patients undergo radiotherapy at some point during the disease trajectory [1,2]. Despite the fact that new conformal technologies have improved radiotherapy treatment, the dose of ionizing radiation delivered to the normal surrounding tissues is still substantial. Therefore, as cure rates for cancer have improved, the number of cancer survivors who experience radiation-induced gastrointestinal symptoms has also increased manyfold [3]. In Europe, roughly one million pelvic-organ cancer survivors suffer from compromised intestinal health due to radiation-induced gastrointestinal symptoms. Acute gastrointestinal symptoms are experienced at 1 to 2 weeks after the commencement of radiotherapy [4]. The acute side effects are usually transient, and some of the symptoms subside within 2–6 weeks from the end of the radiotherapy. Diarrhea and uncomfortable flatulence are the most common acute gastrointestinal side effects experienced by the patients undergoing pelvic radiotherapy [5,6]. Chronic side effects occur months to years or even decades after radiotherapy and are characterized by diarrhea, intestinal dysmotility, fecal incontinence, malabsorption, tenesmus, steatorrhea, urgency, blood discharge, mucus discharge, and excessive production of odorous gases [7,8].

Radiation-induced acute side effects may be due to the disruption of the epithelial barrier, crypt cell death, mucosal inflammation, and the accumulation of inflammatory cells [9,10]. Several studies have shown that ionizing radiation induces the synthesis of various pro-inflammatory and fibrogenic cytokines by several different tissues, including the intestines [11,12,13,14,15,16]. Cytokines regulate the immune system and inflammation via a complex and highly coordinated signaling cascade. Resident immune cells are the first immune cells to respond to irradiation by producing pro-inflammatory cytokines, chemokines, and growth factors [17]. The early immune response is mediated through the secretion of the pro-inflammatory cytokines interleukin (IL)-1, IL-6, and tumor necrosis factor (TNF)-α, which in turn activate the resident immune cells, e.g., macrophages and lymphocytes [18]. Increased levels of IL-1, IL-6, and TNF-α are found in the plasma samples of patients soon after irradiation [19,20]. Chemokines also play important roles in recruiting circulating neutrophils, macrophages, lymphocytes, and eosinophils to the radiation-damaged site [17]. This further enhances the ongoing inflammatory processes. It has been shown that the cascade of pro-inflammatory and pro-fibrotic cytokines that is produced immediately after irradiation may persist for weeks to months until the tissue becomes fibrotic [16,21,22]. Molecular evidence also indicates that a “cytokine cascade” with a distinct temporal pattern exists between the early and late effects of irradiation [16].

The health benefits of dietary fiber have been acknowledged for decades [23,24]. Studies have shown that dietary fiber may increase the number of crypts in the colon, thereby decreasing intestinal atrophy and increasing intestinal mass [25]. Dietary fiber increases the nutritional status of the colonic mucosa by increasing the levels of short-chain fatty acids (SCFAs) produced by the gut microbiota [26]. Dietary fiber has also been shown to decrease the permeability of the intestinal mucus and to decrease the levels of serum pro-inflammatory cytokines [27,28]. Despite the known health benefits of dietary fiber, patients may be advised to follow a low-fiber diet during the course of radiotherapy [29,30]. This advice, which is aimed at reducing the frequency of diarrhea and other acute gastrointestinal symptoms, is not unequivocally evidence-based [30]. Hence, the recommendation made to the patients to lower their fiber intake may be counterproductive and may exacerbate gastrointestinal toxicity in these patients. The underlying pathophysiological processes responsible for the beneficial effects of dietary fibers are not completely understood. Apart from health benefits, fibers from oat bran have also been shown to have immunomodulatory effects [31]. Therefore, we measured serum cytokine levels in an experimental mouse model of pelvic radiotherapy to investigate the influence of a fiber-rich bioprocessed oat bran diet on radiation-induced inflammation.

## 2. Materials and Methods

### 2.1. Animals

All animal experiments were performed on C57BL/6J male mice from Charles River Laboratories (Sulzfeld, Germany), with the dietary intervention starting when the mice were 9 weeks old. All the animals were housed at a constant temperature (20 °C) and humidity (42%). A 12-h light/dark cycle was maintained with free access to food and water. All animal procedures were performed in accordance with the Gothenburg Committee of the Swedish Animal Welfare Agency (Ethics permit no. 1458-2018).

### 2.2. Experimental Design

Mice were fed with diets that contained either 15% fiber (bioprocessed oat bran; High-oat) or 0% fiber (No-fiber) and were either subjected to irradiation or only sedated (controls). The dietary intervention started at 2 weeks before irradiation and continued until 1, 6, and 18 weeks after irradiation (Figure 1A). Thereafter, mice were sacrificed and blood samples were collected (1 week and 18 weeks; *n* = 6 per group at each time point, 6 weeks; *n* = 10 per group, except for the No-fiber control group, in which *n* = 9). Henceforth, the four treatment groups will be referred to as the High-oat irradiated (High-oat irr), High-oat control (High-oat cntl), No-fiber irradiated (No-fiber irr), and No-fiber control (No-fiber cntl). The compositions of the diets, High-oat (15% fiber) and No-fiber (0% fiber), are shown in Figure 1B and Appendix A. The bioprocessed liquid oat bran was prepared according to the European patent #2996492, provided by Glucanova AB (Lund, Sweden) [32]. Basal mixture (TD.160816) was purchased from Envigo Teklad Diets (Madison, WI, USA). The purified corn starch (Cargill’s C*Gel 03401) was provided by Caldic Ingredients Sweden AB (Malmö, Sweden).

### 2.3. Irradiation Procedure

Mice were anesthetized with isoflurane (Isoba vet, MSD Animal Health, Milton Keynes, Buckinghamshire, UK) and placed in a silicone mold. The colorectum was irradiated with a total of 32 Gy delivered in four fractions using a clinical linear accelerator (TrueBeam; Varian Medical Systems Inc., Charlottesville, VA, USA), with 6 MV of nominal photon energy and a dose rate of 5.9 Gy/min. The time interval between each fraction was 12 h. A 3 × 3 cm^2^ radiation field was placed so that the lower quadrant of the field irradiated 1.5 cm of the colorectal region. Maximum care was taken to avoid irradiating the spinal cord, the testicles or the bone-marrow containing femur, so as to avoid any mobility problems, hormonal imbalance, or disruption of the immune system. The distance between the source surface and the skin of the mice was 100 cm. A tissue-equivalent bolus of 5 mm thickness was used to cover the irradiated part of the body, to ensure an even distribution of irradiation throughout the underlying tissue. The dose variation within the target volume was estimated to be ± 5%. The irradiation procedure for each mouse was completed within 5 min. Control animals were anesthetized under the linear accelerator but were not subjected to irradiation.

### 2.4. Serum Preparation

Mice were anesthetized with isoflurane and blood was drawn from the heart by puncturing the ventricle. The blood was allowed to clot at room temperature for 30–45 min, then centrifuged at 1000 × *g* for 15 min at 4 °C. Serum was transferred to a new Eppendorf tube and centrifuged again at 10,000 × *g* at 4 °C, to eliminate any platelets or precipitates, and then transferred to a new tube and stored at −80 °C until further use.

### 2.5. Analyses of Serum Cytokines and Chemokines

The serum levels of cytokines and chemokines in the mice were analyzed using the Bio-Plex Mouse Cytokine 23-Plex Assay (Bio-Rad Laboratories AB, Solna, Sweden). The 23 cytokines and chemokines included in the panel were interleukin (IL)-1α, IL-1β, IL-2, IL-3, IL-4, IL-5, IL-6, IL-9, IL-10, IL-12p40, IL-12p70, IL-13, IL-17A, eotaxin, granulocyte colony-stimulating factor (G-CSF), granulocyte-macrophage colony-stimulating factor (GM-CSF), interferon (IFN)-γ, keratinocyte-derived chemokine (KC), monocyte chemoattractant protein-1 (MCP-1), macrophage inflammatory protein (MIP)-1α, MIP-1β, regulated upon activation normal T-cell expressed and secreted (RANTES), and tumor necrosis factor (TNF)-α. The assay was performed in a 96-well filtration plate supplied with the assay kit. The filter plate was pre-wet with a wash buffer before the addition of the samples. The serum samples were diluted (1:4) with a Bio-Plex sample diluent. After priming the plate with the bead solution, 50 μL of samples and standards were added to the plate and incubated for 30 min at room temperature with shaking. After washing three times, 25 μL of the detection antibody was added and the plate was incubated for 30 min, followed by incubation with streptavidin-phycoerythrin (PE) for 10 min. The plate was washed again three times, after which the beads were resuspended in 125 μL assay buffer. Fluorescence intensity was measured using the Bio-Plex 200 system, with examination under a green laser to determine the cytokine and chemokine concentrations (pg/mL), while the type of cytokine or chemokine was identified by a red laser. The data were analyzed using the Bio-Plex Manager version 6.1 software with 5PL curve fitting. Standard curves for the cytokines were generated using the reference cytokine concentration supplied by the manufacturer. All the standards and samples were run in duplicate.

### 2.6. Statistical Analysis

All the statistical analyses were performed with the GraphPad Prism 8 software (GraphPad Software Inc., San Diego, CA, USA). For the data that were normally distributed, a two-tailed Student’s *t*-test was used to compare the difference between the two groups. If the data were not normally distributed, then a non-parametric (Mann–Whitney) test was used. Extreme outliers were identified using the GraphPad Prism by ROUT method, which is based on nonlinear regression and false discovery rate with Q value set at 2% [33]. A few outliers were found and were eliminated from the analysis. The statistical analysis employed for each experiment is described in detail in the respective figure legends. In the figures, statistically significant differences are indicated with asterisks as follows: * *p* ≤ 0.05; ** *p* ≤ 0.01; *** *p* ≤ 0.001; and **** *p* ≤ 0.0001.

### 2.7. Principal Component Analysis

Principal component analysis (PCA) is a multivariate statistical analysis technique that is used to reduce the dimensionality of large datasets, by transforming several related variables to a smaller set of uncorrelated variables, which are also called principal components [34]. PCA is an unsupervised method that can be used to identify groupings within the multivariate data. A PCA score scatter plot was prepared to visualize the difference between various groups with respect to their cytokine profiles at different time points. A PCA loading scatter plot was prepared to observe the associations between the different groups and the cytokines at different time points using the SIMCA version 15 software (Umetrics, Umeå, Sweden) [35,36].

### 2.8. Ingenuity Pathway Analysis

The bioinformatics software Ingenuity Pathway Analysis (IPA; Qiagen, Sollentuna, Sweden) was used to identify canonical pathways and biological functions or diseases associated with the dataset [37]. This software uses information from the Ingenuity Knowledge Base, which is a repository of molecular interactions and functional annotations created from millions of individually modeled relationships between proteins, genes, complexes, cells, tissues, metabolites, drugs, and diseases, all of which have been manually curated from the peer-reviewed literature. At each time point, the ratios of the average concentrations of each cytokine in the four combinations were determined to compare the effects of the following: (1) radiation on the High-oat groups (High-oat irr vs. High-oat cntl); (2) radiation on the No-fiber groups (No-fiber irr vs. No-fiber cntl); (3) diet on the radiated groups (High-oat irr vs. No-fiber irr); and (4) diet on the non-radiated groups (High-oat cntl vs. No-fiber cntl groups). The data were subsequently log_2_ transformed to obtain the log_2_ fold-change values. The log_2_ fold-change data were then uploaded into IPA, to identify canonical pathways and biological functions or diseases that were associated with irradiation and/or diet within the dataset. The “comparison analysis” function in IPA was used to compare the four combinations of High-oat irr vs. High-oat cntl, No-fiber irr vs. No-fiber cntl, High-oat irr vs. No-fiber irr, and High-oat cntl vs. No-fiber cntl. This strategy was designed to identify canonical pathways and biological functions or diseases that were upregulated or downregulated as a consequence of irradiation and/or diet for each time point.

## 3. Results

### 3.1. Serum Cytokine Levels at 1 Week Post-Irradiation

Serum cytokine levels were analyzed at 1 week post-irradiation to examine the signs of acute inflammation. An effect of the irradiation was observed for the pro-inflammatory cytokine IL-1α, whereby the High-oat irr group and the No-fiber irr group had significantly higher levels of IL-1α than the High-oat cntl group (*p* = 0.026) and No-fiber cntl group, respectively (*p* = 0.009). For the pro-inflammatory cytokine IL-2, the effect of irradiation was observed only in the No-fiber diet, such that the No-fiber irr group had significantly higher levels of IL-2 than the No-fiber cntl group (*p* = 0.039) (Figure 2).

An effect of dietary fiber was observed in mice that were irradiated, in that the High-oat irr group had significantly lower levels of the cytokines IL-12p40, G-CSF, IL-1α, IL-3, and IL-13 than the No-fiber irr group (*p* = 0.036, *p* = 0.002, *p* = 0.009, *p* = 0.037, and *p* = 0.024, respectively) (Figure 2).

An effect of dietary fiber was also observed in mice that were not irradiated, in that the High-oat cntl group had significantly lower levels of the cytokines G-CSF, IL-1α, IL-1β, IL-12p70, and IL-10 (*p* = 0.035, *p* = 0.004, *p* = 0.013, *p* = 0.024, and *p* = 0.017, respectively) and the chemokines KC, MIP-1α, and MIP-1β (*p* = 0.009, *p* = 0.041, and *p* = 0.013, respectively), as compared to the No-fiber cntl group (Figure 2). No significant differences were observed between the groups in terms of the levels of IL-5, IL-6, IL-17, GM-CSF, IFN-γ, TNF-α, eotaxin, MCP-1, RANTES, IL-4, and IL-9 (Appendix A).

A PCA score scatter plot was generated to identify the similarities and differences between the various groups with respect to cytokine profiles. At 1 week post-irradiation, the No-fiber irr and No-fiber cntl groups were highly correlated to each other with respect to cytokine levels, and therefore clustered together in the left quadrants of the PCA plot, with some exceptions. Similarly, the High-oat irr and High-oat cntl groups had similar cytokine levels but different cytokine profiles compared to the No-fiber irr and No-fiber cntl groups, and therefore were segregated from the latter two groups in the right quadrants of the PCA plot (Figure 3A). A PCA loading scatter plot was generated to observe the associations between the various groups and the cytokines. We observed that at 1 week post-irradiation, all the cytokines were directly or positively associated with the No-fiber irr and No-fiber cntl groups, whereas the High-oat irr and High-oat cntl groups were inversely associated with all the cytokines (Figure 3B).

### 3.2. Serum Cytokine Levels at 6 Weeks Post-Irradiation

Serum cytokine levels were analyzed at 6 weeks post-irradiation to observe the intermediate inflammation. The effect of irradiation was observed only in the mice fed the No-fiber diet, where the No-fiber irr group had significantly higher levels of IL-12p40 and G-CSF than the No-fiber cntl group (*p* = 0.016 and *p* = 0.017, respectively). An effect of dietary fiber was observed in mice that were irradiated, in that the High-oat irr group had significantly lower levels of IL-12p40 than the No-fiber irr group (*p* = 0.038) (Figure 4).

There was no significant effect of dietary fiber between the non-irradiated control groups, i.e., the High-oat cntl and No-fiber cntl groups. Thus, the levels of IL-1α, IL-1β, IL-2, IL-3, IL-5, IL-6, IL-12p70, IL-17, GM-CSF, IFN-γ, TNF-α, KC, MIP-1α, MIP-1β, eotaxin, MCP-1, RANTES, IL-4, IL-10, and IL-13 were very similar (Appendix A). The concentration of IL-9 was below the detection limit for almost all the samples, so it was excluded from the analysis.

The PCA score scatter plot reveals that, at 6 weeks post-irradiation, the groups were segregated according to irradiation rather than a diet. As a consequence, the High-oat irr and No-fiber irr groups were more abundant in the upper quadrants of the PCA plot, whereas the High-oat cntl and No-fiber cntl groups were more abundant in the lower quadrants of the PCA plot (Figure 5A). The PCA loading scatter plot showed that the No-fiber irr and High-oat cntl groups were directly associated with all the cytokines, whereas the No-fiber cntl and High-oat irr groups were inversely associated with all the cytokines (Figure 5B).

### 3.3. Serum Cytokine Levels at 18 Weeks Post-Irradiation

Serum cytokine levels were analyzed at 18 weeks post-irradiation to observe the signs of chronic inflammation. An effect of irradiation was observed for the cytokine GM-CSF, where the No-fiber irr group had close to significantly higher levels of GM-CSF than the No-fiber cntl group (*p* = 0.057) (Figure 6).

An effect of dietary fiber was observed in mice that were irradiated, in that the High-oat irr group had significantly lower levels of the cytokines IL-12p40, IL-1β, IL-2, IL-3, GM-CSF, IFN-γ, IL-10, IL-4, and IL-9 (*p* = 0.026, *p* = 0.011, *p* = 0.004, *p* = 0.026, *p* = 0.024, *p* = 0.004, *p* = 0.009, *p* = 0.017, and *p* = 0.009, respectively) and the chemokines KC, MIP-1α, eotaxin, and MCP-1 (*p* = 0.015, *p* = 0.002, *p* = 0.002, and *p* = 0.004, respectively), as compared to the No-fiber irr group (Figure 6).

An effect of dietary fiber was also observed in mice that were not irradiated, whereby the High-oat cntl mice had significantly lower levels of the pro-inflammatory cytokines IL-1α, IL-12p70, and IL-17 (*p* = 0.037, *p* = 0.046, and *p* = 0.009, respectively) and the chemokines KC, MIP-1α, MIP-1β, and MCP-1 (*p* = 0.004, *p* = 0.041, *p* = 0.033, and *p* = 0.002, respectively), as compared to the No-fiber cntl mice (Figure 6). No significant differences were observed between any of the groups for the cytokines IL-5, IL-6, G-CSF, TNF-α, RANTES, and IL-13 (Appendix A).

The PCA score scatter plot showed that, at 18 weeks post-irradiation, the High-oat irr and High-oat cntl groups were highly correlated to each other with respect to cytokine levels and clustered together in the right quadrants of the PCA plot. Similarly, the No-fiber irr and No-fiber cntl groups had similar cytokine profiles that were distinct from those of the High-oat irr and High-oat cntl groups, and therefore segregated in the left quadrants of the PCA plot, with few exceptions (Figure 7A). The PCA loading scatter plot revealed that the High-oat irr and High-oat cntl groups were directly associated with IL-5 and inversely associated with the remaining cytokines analyzed, whereas the No-fiber irr and No-fiber cntl groups were directly associated with all the cytokines, with the exception of IL-5 (Figure 7B).

### 3.4. Ingenuity Pathway Analysis of the Cytokine Profiles

IPA was used to identify the canonical pathways and biological functions or diseases associated with the cytokine profiles. Using IPA, we examined the relationships between the cytokine profiles and the canonical pathways at different time points. The most significant top 15 canonical pathways at each time point are shown in Appendix A. The 1- and 18-week time points shared similar canonical pathways, whereas the 6-week time point shared 14 out of 15 canonical pathways with the 1- and 18-week time points. At 1 and 18 weeks post-irradiation, the T-helper cell type 1 (Th1) and Th2 activation pathways were strongly implicated, whereas at 6 weeks post-irradiation the Th17 activation pathway was heavily involved. The data from all the time points reveal that the intestinal epithelial cells, macrophages, and T-helper cells were likely strongly activated by IL-17A and IL-17F to release other cytokines and chemokines. Similarly, the relationships between the cytokines and the top 15 biological functions or diseases were studied at each time point (Appendix A). All the time points shared similar biological functions or diseases. The high degrees of enrichment for the categories of immune cell trafficking, inflammatory response, cell-mediated immune response, and inflammatory disease suggest that there is ongoing inflammation. The functional enrichments in terms of tissue development, tissue morphology, cellular development, cellular growth and proliferation, and organismal injury and abnormalities suggest substantial changes in histomorphology.

To observe the effects of irradiation, the comparison analysis tool in IPA was used to compare the High-oat irr vs. High-oat cntl and No-fiber irr vs. No-fiber cntl groups. Similarly, the High-oat irr vs. No-fiber irr and High-oat cntl vs. No-fiber cntl groups were compared to observe the dietary fiber effects. A comparison analysis tool was used to examine the relationships between the canonical pathways and biological functions or diseases associated with the compared groups. At the 1- and 18-week time points, a clear effect of irradiation was observed, as most of the canonical pathways and biological functions were upregulated by irradiation (indicated by the orange coloration in Figure 8A,C and Figure 9A,C). However, most of the canonical pathways and biological functions that were upregulated by irradiation were downregulated in the mice that were fed the High-oat diet, as compared to the No-fiber diet (indicated by the blue coloration in Figure 8A,C and Figure 9A,C). This indicates that the High-oat diet protects against the harmful effects of irradiation, when compared with the No-fiber diet. At the 6-week time point, a clear effect of irradiation was observed only in mice fed with the No-fiber diet as most of the canonical pathways and biological functions were upregulated by irradiation (indicated by the orange coloration in Figure 8B and Figure 9B).

## 4. Discussion

Preclinical and clinical studies have shown that pelvic radiotherapy increases cytokine levels soon after irradiation, and that this plays an important role in radiation-induced gastrointestinal toxicity [38,39,40]. However, long-term studies of post-irradiation cytokine profiles are scarce. The main goal of the present study was to evaluate whether a fiber-rich bioprocessed oat bran diet can modify radiation-induced inflammation. We show that a No-fiber diet generates an abundance of circulating pro-inflammatory cytokines for at least 18 weeks after the irradiation, indicating an exacerbation of radiation-induced inflammation. In contrast, a fiber-rich bioprocessed oat bran diet causes decreased levels of pro-inflammatory cytokines, indicating the mitigation of radiation-induced inflammation. Our data also indicate that radiation-induced inflammation is a long-term phenomenon that, at least in mice, persists for several months without resolving.

With the advent of new technologies, a high dose of radiation can be delivered to the tumor site. Our experimental model of pelvic radiotherapy is unique in that we irradiated the mice with the same clinical linear accelerator (LINAC) that is used to irradiate cancer patients [41]. Thus, high-energy photons can be delivered to a highly defined area, thereby avoiding undesirable irradiation of non-target organs. Our previous study showed that irradiation with 8 Gy × 4 fractions was best suited for our model as it inflicted similar pathologic changes in the mouse intestinal mucosa, notably crypt degeneration, as seen in a cancer survivor’s intestinal mucosa irradiated at the pelvic region [41,42]. Furthermore, these mice appear to maintain their overall health and a normal lifespan despite having received high doses of irradiation. Therefore, our model is suitable for studying the long-term effects of irradiation, given that radiation-induced symptoms in patients may persist for decades after the initial radiotherapy [43,44].

At all-time points, the metrics concerning cytokines and chemokines are consistent with the notion of an irradiation effect. At 1 week post-irradiation, increased levels of IL-1α were seen in the mice subjected to irradiation, as compared to their respective control group. Similar results have been shown in previous studies, where increased levels of IL-1α have been reported in both mice and rats subjected to irradiation [9,16]. IL-1α is a pro-inflammatory cytokine that is mainly produced by activated macrophages, neutrophils, epithelial cells, and endothelial cells. It is also involved in the acute-phase response to radiation-induced injury and plays an important role in tissue remodeling by inducing fibrosis, as IL-1α has key activities in the regulation of fibroblast proliferation and connective tissue production [20,45]. Irradiation also caused increased IL-2 levels in the mice fed the No-fiber diet. Increased levels of IL-2 have been reported in patients with radiation-induced proctitis [46]. IL-2 is secreted by activated T lymphocytes and is involved in T-helper cell proliferation. The increased levels of IL-2 in the No-fiber irradiated mice are an indication that the cytotoxic T cell-mediated immune mechanism may have a role in the pathogenesis of radiation-induced inflammation.

At 6 weeks post-irradiation, increased levels of G-CSF were observed in the irradiated mice fed a No-fiber diet, as compared to the non-irradiated control mice. G-CSF is a potent hematopoietic factor that enhances the survival and differentiation of myeloid lineage cells [47]. It is known to act in the generation, mobilization, and function of neutrophils, which are key innate immune cells that protect against invading microbes [48,49,50]. In accordance with our results, the levels of G-CSF have been found to be increased during radiotherapy in patients with prostate cancer [51].

At 18 weeks post-irradiation, increased levels of GM-CSF were seen in the mice fed the No-fiber diet and subjected to irradiation, as compared to the control mice. Similarly, increased levels of IL-1α, IL-3, IL-12p40, IL-12p70, MIP-1β, and TNF-α were observed in both the irradiated groups, as compared to their respective control groups. These findings indicate that radiation-induced inflammation is a long-term process, contradicting the notion that radiation-induced inflammation is an acute process that gets resolved within a short period [40].

The health benefits of oat bran have been recognized for decades [52,53]. The bioprocessed oat bran used in the present study contains 52% fiber, of which 28% is β-glucan having a molecular weight of around 100 kDa. Beta-glucan can modulate both adaptive and innate immunity. Beta-glucan is a potent activator of the innate immune system including macrophages, neutrophils and cytotoxic lymphocytes such as natural killer (NK) cells. Oat beta-glucan enhances resistance towards various bacterial, protozoal, and viral infections by its ability to activate macrophages, thereby enhancing host immune defense [54,55]. Dietary fiber may alter the microbiota composition and increase the production of beneficial short-chain fatty acids (SCFAs), such as acetate, propionate, and butyrate. Butyrate is known to exert anti-inflammatory effects through the modulation of NF-κB activation [56]. Apart from dietary fiber, oat bran also contains a large variety of bioactive components, including phenolic acids and avenanthramides, which may also contribute to its physiological effects.

Mice fed with a fiber-rich bioprocessed oat bran diet had lower levels of cytokines and chemokines compared to the mice fed a No-fiber diet, even in the absence of ionizing irradiation. The effects of dietary fiber on the non-irradiated animals were examined at the 1-week time point. The High-oat control group had lower levels of pro-inflammatory cytokines and chemokines, such as IL-1β, IL-12p70, KC, MIP-1α, and MIP-1β, and the anti-inflammatory cytokine IL-10, as compared to the No-fiber control group. Oat bran has been previously shown to decrease the levels of IL-1β, KC, MIP-1α, and IL-10 [57,58,59,60]. Oat beta-glucan has also been previously shown to decrease the mRNA and protein expression of pro-inflammatory cytokines such as IL-1β, IL-6, and TNF-α in mice suffering from dextran sulfate sodium (DSS)-induced ulcerative colitis [57]. IL-12p70, mainly produced by dendritic cells, macrophages, neutrophils, and B cells, is important for the growth and differentiation of B and T cells, as well as in the proliferation of natural killer (NK) cells. The IL-12 level is an indicator of the intensity of colonic inflammation [61]. KC is predominantly secreted by macrophages, neutrophils, and epithelial cells and has neutrophil chemoattractant activity. MIP-1α is a chemoattractant for macrophages, monocytes, and neutrophils, while MIP-1β attracts NK cells, monocytes, and a variety of immune cells. IL-10, which is produced by Th2 cells, cluster of differentiation (CD) 8+ T cells and B cells, and macrophages, is a potently immunosuppressive cytokine, repressing the expression of cytokines, such as TNF-α, IL-6, and IL-1 by activated macrophages [62].

At 6 weeks post-irradiation, there were no clear differences in cytokine levels between the non-irradiated control groups. However, at 18 weeks post-irradiation, the High-oat control group showed decreased levels of IL-17 and MCP-1, when compared to the No-fiber control group. IL-17 is mainly produced by Th17 cells and is involved in both acute and chronic inflammation. Patients with inflammatory bowel disease (IBD) have increased levels of IL-17 [63]. High-fiber diet has previously been shown to decrease the serum IL-17 levels in a murine model of autoimmune hepatitis [64]. MCP-1 is a chemokine that promotes the recruitment of immune cells, such as monocytes, T cells, and dendritic cells, to sites of inflammation that appear following either tissue injury or infection. Wheat arabinoxylan has been previously shown to decrease the production of MCP-1 by dendritic cells [59].

Bioprocessed oat bran diet decreased the levels of cytokines in the irradiated mice at all time points. At 1 week post-irradiation, the High-oat irradiated group had lower levels of IL-3 and IL-13, as compared to the No-fiber irradiated group. IL-3 is produced by activated T cells and basophils. The level of IL-3 has been correlated with fatigue in patients with prostate cancer who were receiving radiotherapy [65]. IL-13, which is secreted mainly by Th2 cells, is upregulated by irradiation and is a key factor in the progression of radiation-induced fibrosis [66,67].

At 6 weeks post-irradiation, the High-oat irradiated group had lower levels of IL-12p40 than the No-fiber irradiated group. The main source of IL-12p40 is T cells, and it activates both T and NK cells. At 18 weeks post-irradiation, the High-oat irradiated group had lower levels of IL-1β, GM-CSF, IFN-γ, eotaxin, IL-4, IL-9, and IL-10, as compared to the No-fiber irradiated group. IL-1β is secreted primarily by monocytes and macrophages, as well as by non-immune cells, such as fibroblasts and endothelial cells. Several studies have shown that IL-1β is a major mediator of radiation-induced intestinal damage, and that the administration of an IL-1 receptor antagonist leads to reduced intestinal injury [68,69]. IL-1β is also responsible for exacerbating mucositis by causing the disruption of tight junctions in the intestines of mice [70]. GM-CSF is secreted by macrophages, T cells, mast cells, NK cells, endothelial cells, and fibroblasts. The level of serum GM-CSF has been shown to increase post-irradiation [71]. IFN-γ is mainly produced by NK cells and Th1 cells as part of the innate immune response. Irradiation increases the levels of IFN-γ, whereas cereal fiber has been shown to decrease IFN-γ levels [72,73]. Eotaxin is a chemokine that is responsible for the chemotaxis of eosinophils. Abdominal irradiation has been shown to increase the expression of eotaxin, which plays a critical role in radiation-induced fibrosis [74]. IL-4 is a pleiotropic cytokine that is mainly produced by activated Th2 cells, mast cells, and basophils. It has been shown that the levels of IL-4 increases post-irradiation in patients with prostate cancer, and has been implicated in radiation-induced fibrosis [75,76]. Similar to our results, cereal fiber has been shown to decrease the levels of IL-4 in the sera of healthy persons [73]. Arabinoxylan extracted from rice bran has also been shown to inhibit the production of IL-4 by bone marrow-derived mast cells [77]. IL-9 is predominantly secreted by a subset of CD4+ T cells called Th9 cells and is involved in the pathogenesis of IBD. It causes disruption of the intestinal barrier, thereby enhancing bacterial translocation into the mucosa, and inhibits intestinal wound healing [78].

Overall, the levels of 5/23, 1/23, and 13/23 cytokines were found to be significantly lower in the High-oat irradiated group than in the No-fiber irradiated group at 1, 6, and 18 weeks post-irradiation, respectively. These results indicate that a fiber-rich diet helps to decrease radiation-induced inflammation not only acutely, but also in the longer term, and thereby might help in reducing late radiation-induced toxicity. Moreover, we also show that a lack of dietary fiber increases the levels of pro-inflammatory cytokines in animals that have not been subjected to irradiation. This is important when one considers that patients with already aggravated inflammatory states are at extremely high risk of developing severe acute and late toxicities after pelvic radiotherapy [79]. Thus, reducing an already existing inflammatory state through dietary interventions is of potential interest. In addition, the chronic effects of inflammation are associated with high morbidity and mortality in patients receiving radiotherapy. Similar results emerged from our IPA analysis, in which the canonical pathways and biological functions or diseases that are upregulated by irradiation appear to be downregulated by the High-oat diet, as compared to the No-fiber diet. The IPA data from all the time points reveal that the oat bran diet downregulated high mobility group box 1 (HMGB1) and triggering receptor expressed on myeloid cells-1 (TREM1) signaling pathways compared to No-fiber diet after irradiation. Both HMGB1 and TREM1 signaling pathways have been shown to promote inflammatory responses. HMGB1 acts as an inducer to activate macrophages and leukocytes and promotes the production of inflammatory cytokines such as IFN-γ, TNF-α, and IL-6 [80]. TREM1 can activate myeloid cells to release inflammatory cytokines including TNF-α, IL-1β, and IL-6 [81]. Taken together, our mouse model data indicate that an oat bran-containing diet protects against the harmful effects of irradiation by reducing radiation-induced inflammation, which in contrast may be exacerbated by the No-fiber diet.

## 5. Conclusions

Elevated serum cytokine levels were found in mice at 1, 6, and 18 weeks after irradiation, indicating that radiation-induced inflammation is a long-term phenomenon. A diet without fiber increased the production of pro-inflammatory cytokines for many months after the irradiation. In contrast, a fiber-rich oat bran diet resulted in decreased cytokine levels, thereby appearing to reduce the acute, intermediate, and chronic radiation-induced inflammation. Moreover, the fiber-rich diet appeared to downregulate the canonical pathways and biological functions or diseases that were upregulated by the irradiation, as compared to the fiber-free diet. Therefore, a diet high in fiber could help to reduce radiation-induced inflammation, whereas the absence of it may exacerbate radiation-induced inflammation. Based on these results in mice, it could be advisable to conduct clinical dietary interventions to confirm if patients undergoing radiotherapy would also benefit from a high-fiber diet during their treatment.

## Figures and Tables

**Figure 1 nutrients-12-02172-f001:**
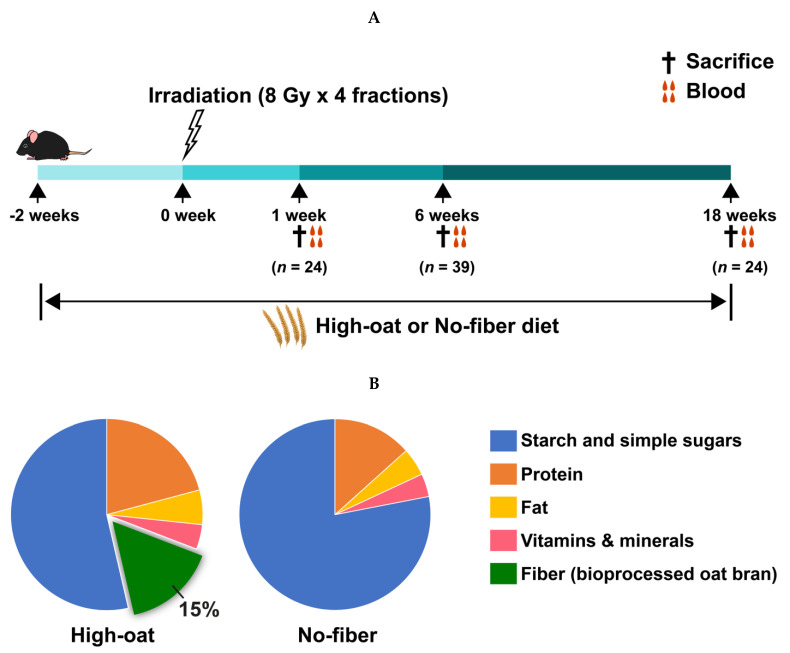
Experimental design. (**A**) Mice were fed the High-oat or No-fiber diet starting at 2 weeks before irradiation until 1, 6, and 18, weeks post-irradiation. Mice were sacrificed at 1, 6, and 18 weeks post-irradiation followed by blood sample collection. (**B**) Compositions of the High-oat and the No-fiber diets.

**Figure 2 nutrients-12-02172-f002:**
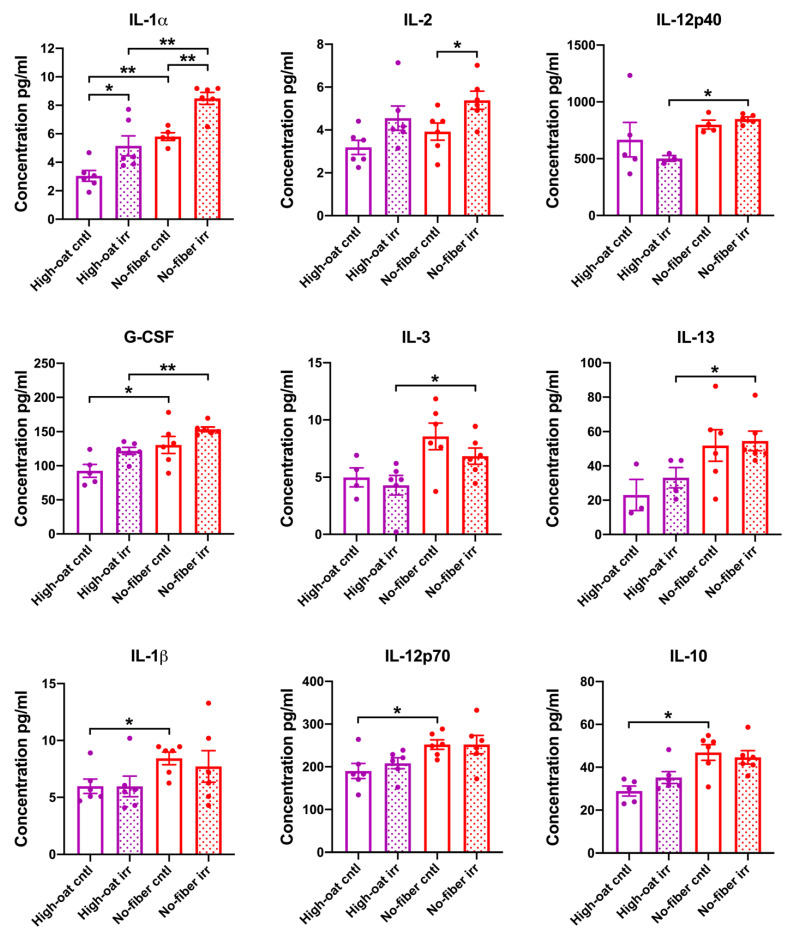
Serum cytokine and chemokine levels in mice at 1 week post-irradiation. A two-tailed Mann–Whitney test was used to compare the cytokine and chemokine levels in High-oat irr vs. High-oat cntl, No-fiber irr vs. No-fiber cntl, High-oat irr vs. No-fiber irr, and High-oat cntl vs. No-fiber cntl groups. Data shown are average concentrations and the error bars represent SEM. * *p* ≤ 0.05; and ** *p* ≤ 0.01.

**Figure 3 nutrients-12-02172-f003:**
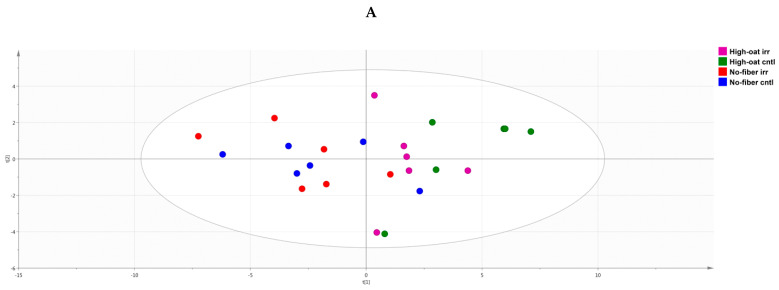
Principal component analysis (PCA), a multivariate dimensionality reduction analysis was performed on serum cytokines and chemokines levels in mice at 1 week post-irradiation. (**A**) PCA score scatter plot of cytokine profiles, showing discrimination between the groups. (**B**) PCA loading scatter plot showing the associations between the cytokines and different groups.

**Figure 4 nutrients-12-02172-f004:**
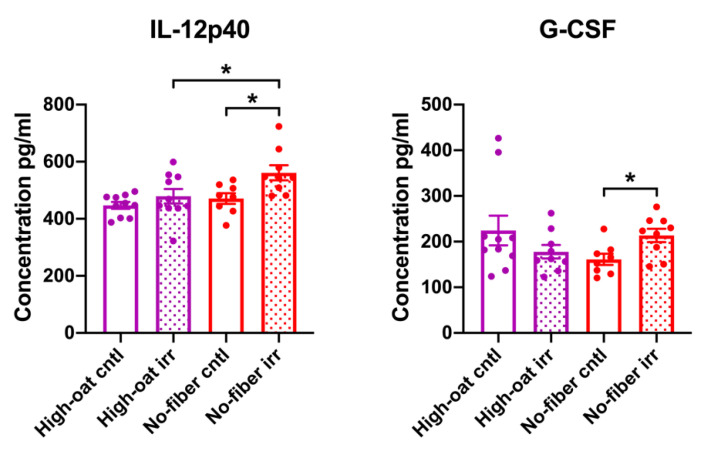
Serum cytokine and chemokine levels in mice at 6 weeks post-irradiation. A two-tailed Student’s *t*-test was used to compare the cytokine levels in the High-oat irr vs. High-oat cntl, No-fiber irr vs. No-fiber cntl, High-oat irr vs. No-fiber irr, and High-oat cntl vs. No-fiber cntl groups. Data shown are average concentrations and the error bars represent SEM. * *p* ≤ 0.05.

**Figure 5 nutrients-12-02172-f005:**
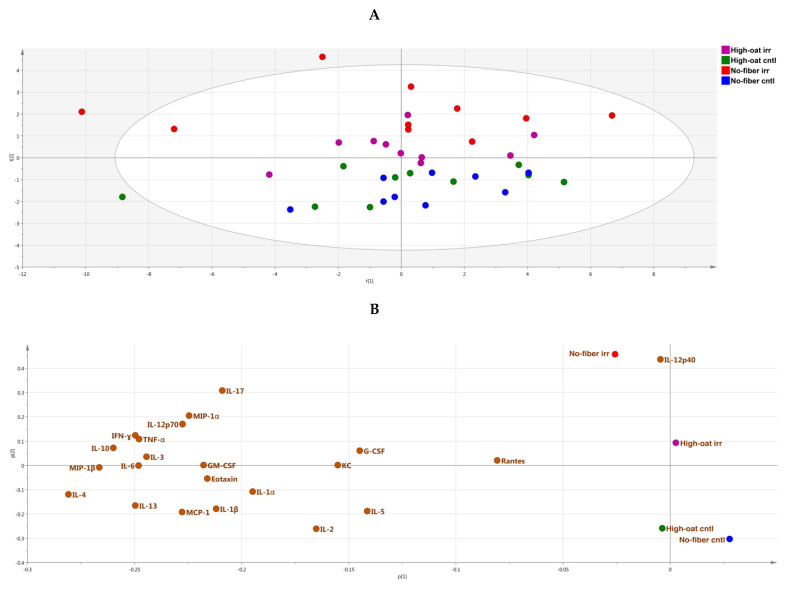
PCA analysis of serum cytokines and chemokines levels in mice at 6 weeks post-irradiation. (**A**) PCA score scatter plot of cytokine profiles, showing discrimination between the groups. (**B**) PCA loading scatter plot showing the associations between the cytokines and different groups.

**Figure 6 nutrients-12-02172-f006:**
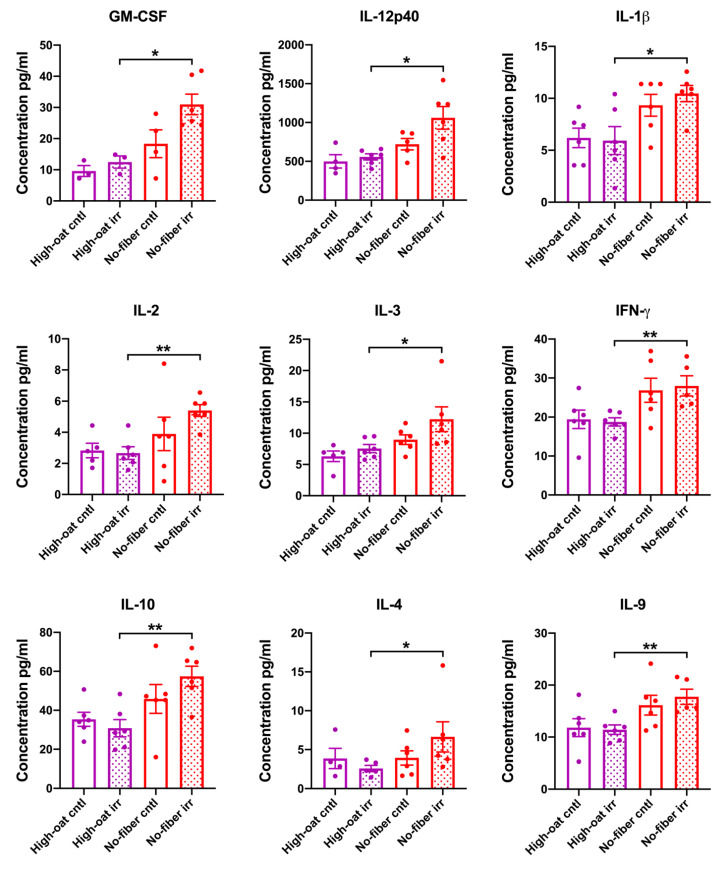
Serum cytokine and chemokine levels in mice at 18 weeks post-irradiation. A two-tailed Mann–Whitney test was used to compare the cytokine and chemokine levels in High-oat irr vs. High-oat cntl, No-fiber irr vs. No-fiber cntl, High-oat irr vs. No-fiber irr, and High-oat cntl vs. No-fiber cntl groups. Data shown are average concentrations and the error bars represent SEM. * *p* ≤ 0.05; and ** *p* ≤ 0.01.

**Figure 7 nutrients-12-02172-f007:**
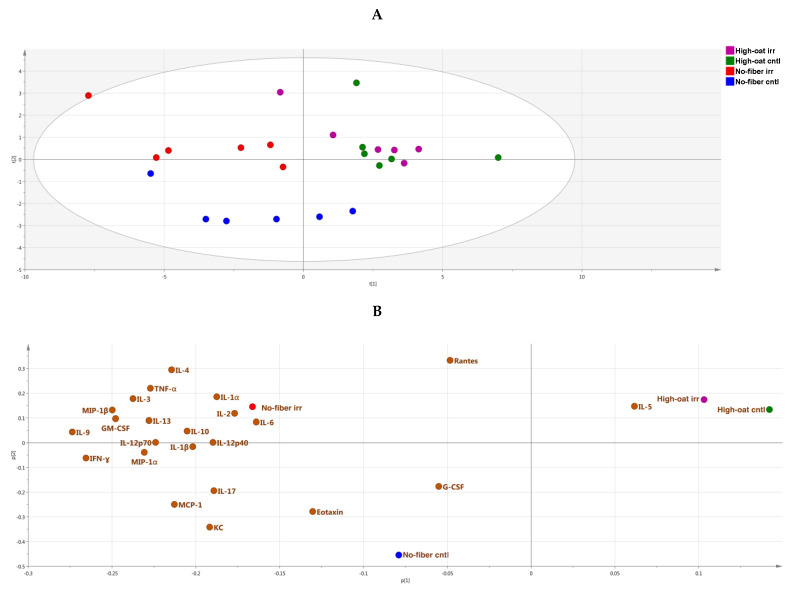
PCA analysis of serum cytokines and chemokines levels in mice at 18 weeks post-irradiation. (**A**) PCA score scatter plot of cytokine profiles, showing discrimination between the groups. (**B**) PCA loading scatter plot showing the associations between the cytokines and different groups.

**Figure 8 nutrients-12-02172-f008:**
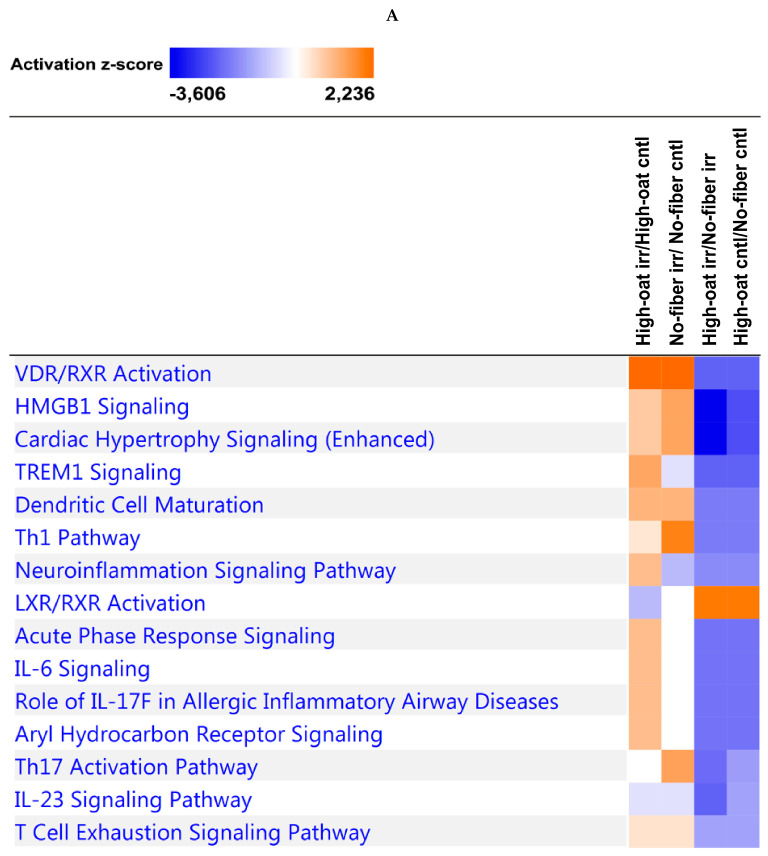
Pathway enrichment analysis. The comparison analysis was used to compare the High-oat irr vs. High-oat cntl, No-fiber irr vs. No-fiber cntl, High-oat irr vs. No-fiber irr, and High-oat cntl vs. No-fiber cntl groups, to observe the top 15 canonical pathways associated with the cytokine profiles at (**A**) 1, (**B**) 6, and (**C**) 18 weeks post-irradiation, ordered according to the highest activation z-score. The orange colorations indicate upregulation of the pathway and the blue colorations indicate downregulation of the pathway.

**Figure 9 nutrients-12-02172-f009:**
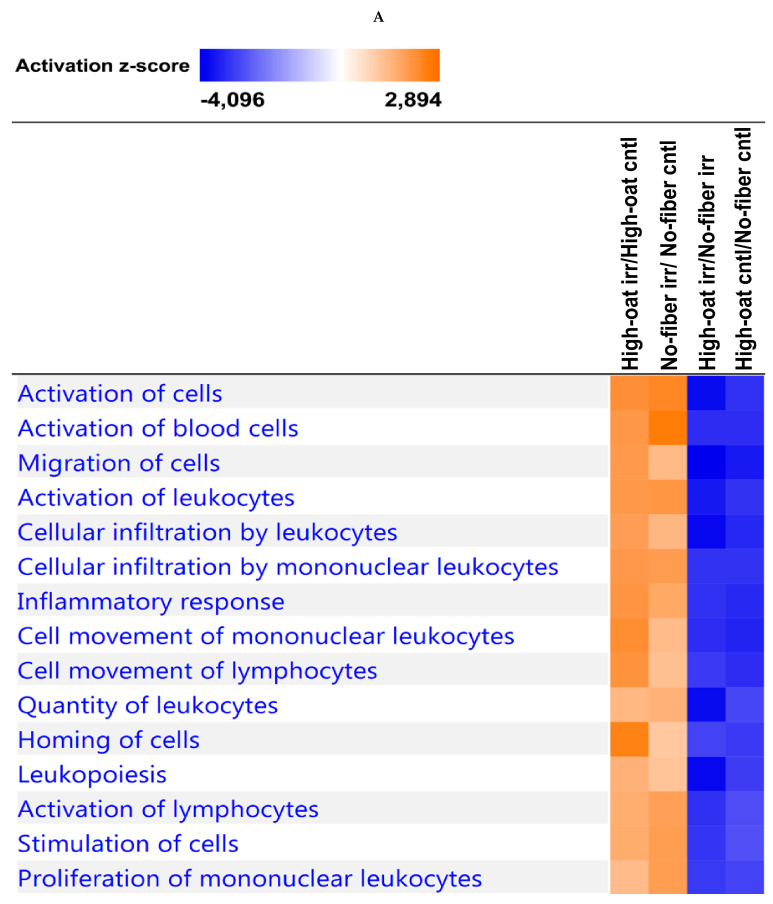
Biological function and disease enrichment analysis. The comparison analysis was used to compare the High-oat irr vs. High-oat cntl, No-fiber irr vs. No-fiber cntl, High-oat irr vs. No-fiber irr, and High-oat cntl vs. No-fiber cntl groups, to observe the top 15 biological functions or diseases associated with the cytokine profiles at (**A**) 1, (**B**) 6, and (**C**) 18 weeks post-irradiation, ordered according to the highest activation z-score. The orange colorations indicate activation of the functions or diseases and the blue colorations indicate inhibition of the functions or diseases.

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
