# Peer review of "Dietary Oat Bran Reduces Systemic Inflammation in Mice Subjected to Pelvic Irradiation"

_nutrients, 2020, doi:10.3390/nu12082172_

Round 1

Reviewer 1 Report

The manuscript submitted for publishing in Nutrients presents interesting results however needs major revision.

  1. There are some doubts arising from group codes. It seems that the manuscript presents some of the results from the wider experiment because there is no need for coding High oat fiber and No fiber when there is only one level of fiber.
  2. Also the manuscript theme is far too general - the study only uses oat bran fiber - which is particular in case of containing beta-glucan.
  3. There are some doubts about the ingredient called "oat bran fiber" - Authors are indicating the patent number of Glucanova but the mentioned patent referred to "METHOD FOR PREPARING A LIQUID OAT BASE AND PRODUCTS PREPARED BY THE METHOD" and the resulting product is not actually fiber.
  4. So there are strong methodological errors at the beginning which influence the correctness of the conducted discussion part.
  5. The impact of so composed diet can't only be discussed in perspective of fiber content - as the used ingredient is not a fiber  - at least according to cited patent.
  6. The discussion should be more focused on bioactive components of such ingredient which are mainly oat beta-glucan fraction of specific molar mass as well as arabinoxylans.
  7. The manuscript need profound re-writting starting from the analysis of an ingredient called "fiber" and its characteristics.

Reviewer 2 Report

In the present manuscript “Oat bran fiber reduces systemic inflammation in mice subjected to pelvic irradiation” the authors explore the influence of dietary fiber on radiation-induced inflammation. This understanding could help to guide physicians in the dietary advises gave to patients undergoing radiotherapy to treat pelvic-organ cancer.

  1. In general, the submitted manuscript is very well structured and with careful writing, without major spelling or grammatical errors.
  2. Highlight phrases and a summary in the form of a graphical abstract will help the readers to easily understand the objective of this work.
  3. Introduction provides important theoretical considerations of the different topics necessary for the all understanding of the paper.
  4. Methodology provides enough details and is clearly described. Statistical analysis was properly done.
  5. Which was the rational for the irradiation protocol? Compared to the clinics, the fraction doses are very high. An explanation for the chosen irradiation protocol must be provide.
  6. Results are clearly presented, however, given the nature of the results, vertical scatter plots would be preferable to give more detail on the serum cytokine and chemokine levels distribution and variance compared to column bar graphs.
  7. Discussion presents many findings of other authors corroborating the original results presented in this manuscript. Results are strictly discussed from the immunomodulatory effects of radiation alone to the influence of fiber-rich bioprocessed oat bran diet on radiation-induced inflammation.
  8. Although the discussion addresses in great detail the differences in cytokines/chemokines levels, it seems it lacks from a more detailed discussion on the IPA analysis.

Round 2

Reviewer 1 Report

The authors have corrected the manuscript as recommended. Currently it does not raise any major objections and can be accepted for publishing.